# PCR for detection of *Leishmania donovani* from microscopically negative tissue smears of suspected patients in Gondar, Ethiopia

**Roma Melkamu**[1,2]*, **Nega Berhane**[2], **Bart K. M. Jacobs**[3], **Rezika Mohammed**[1], **Mekibib Kassa**[1], **Arega Yeshanew**[1], **Helina Fikre**[1], **Saba Atnafu**[1], **Saskia van Henten**[3], **Johan van Griensven**[3], **Myrthe Pareyn**[3]

**1** Leishmaniasis Research and Treatment Center, University of Gondar, Gondar, Ethiopia, **2** Institute of Biotechnology, University of Gondar, Gondar, Ethiopia, **3** Clinical Sciences Department, Institute of Tropical Medicine, Antwerp, Belgium

* roma.melkamu@yahoo.com

## Abstract

### Background

As untreated visceral leishmaniasis (VL) is fatal, reliable diagnostics are pivotal for accurate treatment allocation. The current diagnostic algorithm for VL in Ethiopia, which is based on the rK39 rapid diagnostic test and microscopy of tissue smears, lacks sensitivity. This probably leads to missed cases and patients not receiving treatment.

### Methodology

We conducted a retrospective study on stored microscopically negative spleen and bone marrow smears from suspected VL patients collected at the Leishmaniasis Research and Treatment Center (LRTC) in Gondar, northern Ethiopia between June 2019 and November 2020. Sociodemographic, clinical and treatment data were collected and samples were tested by real-time PCR targeting kinetoplast DNA.

### Principle findings

Among the 191 eligible samples (135 spleen and 56 bone marrow) with a microscopically negative and valid PCR result, 119 (62.3%) were positive by PCR, although Ct values for some were high (median 33.0). Approximately three quarters of these undiagnosed primary VL (77.3%) and relapse (69.6%) patients did not receive antileishmanial treatment. Of the 56 microscopically negative bone marrow samples, 46 (82.1%) were PCR positive, which is considerably higher compared to the microscopically negative spleen samples, for which 73 out of 135 (54.1%) were PCR positive. The odds of being PCR positive were significantly higher for bone marrow aspirates and higher when white blood cell values were lower and splenomegaly (in cm) was more pronounced.

**Data Availability Statement:** Data will not be made openly accessible due to ethical and privacy concerns. Data can however be made available

after approval of a motivated and written request to ITMs Research Data Access Committee (ITMresearchdataaccess@itg.be).

**Funding:** This work was supported by the Directorate-General Development cooperation and Humanitarian Aid (DGD), under the FA4 framework collaboration of the Institute of Tropical Medicine (Antwerp, Belgium) and the University of Gondar (Gondar, Ethiopia), granted to both JvG and RM. The funders had no role in the study design, data collection and analysis, decision to publish, or preparation of the manuscript.

**Competing interests:** The authors have declared that no competing interests exist.

## Conclusions

This study demonstrates that a lot of suspected VL patients remain undiagnosed and untreated. This indicates the urgent need for better diagnostics for VL in the East-African region. The outcomes of PCR positive should be closely monitored and treatment should be provided if the patient deteriorates. In resource limited settings, implementation of PCR on bone marrow aspirate smears of patients with low WBC values and splenomegaly could lead to considerable improvements in patient management.

## Author summary

As untreated visceral leishmaniasis (VL) is fatal, reliable diagnostics are important for accurate treatment allocation. The current diagnostic algorithm for VL in Ethiopia, which is based on the rK39 rapid diagnostic test and microscopy of tissue smears, lacks sensitivity. This probably leads to missed cases and patients not receiving treatment. To prove this, we conducted a study on stored microscopically negative spleen and bone marrow aspirate smears from suspected VL patients in Gondar, Ethiopia. Clinical and treatment data were collected and samples were tested for *Leishmania* by PCR. We found that about 60% of these microscopically negative samples were PCR positive. This PCR positivity rate was considerably higher in patients with a microscopically negative bone marrow compared to splenic aspirate. Importantly, more than three quarters of the patients with a PCR positive sample was not treated for VL. Overall, our study demonstrates the gap in the diagnostic algorithm for VL in northern Ethiopia, especially when bone marrow samples are used. In resource limited settings, we advise to challenge the current diagnostic algorithm and implement molecular tools to accurately diagnose patients. This could lead to considerable improvements in patient management in Ethiopia and beyond.

## Introduction

Visceral leishmaniasis (VL) is a major public health problem in Ethiopia, with the most important foci situated in the lowlands in the northwest and south of the country [1]. It is caused by an infection with *Leishmania donovani* through the bite of phlebotomine sand flies, and results in an estimated annual incidence of 2,000 to 4,500 cases countrywide [2]. In immunocompetent individuals, *L. donovani* infections often remain asymptomatic. When the infection evolves to active disease, however, it is characterized by fatigue, hepatosplenomegaly, lymphadenopathy, weight loss, progressive fever and pancytopenia. If left untreated, VL is fatal in 95% of the cases [3]. On the contrary, overtreatment should also be avoided, as treatment comes with costs implications and adverse effects. Hence, reliable diagnostic methods are pivotal to guide clinical decision making for treatment of VL patients.

According to the WHO and Ethiopian guidelines, when a VL suspected patient meets the case definition (fever for more than two weeks, splenomegaly and/or lymphadenopathy, either loss of weight, anemia or leukopenia and living in or travel history to a known VL endemic area), a rK39 rapid diagnostic test (RDT) should be performed, and patients should be treated if positive [4,5]. If the rK39 RDT is negative, it should be followed by the direct agglutination test (DAT) or microscopic examination of a Giemsa-stained tissue aspiration to guide the decision to treat. In practice, DAT is rarely available and in three quarters of the cases, tissue

aspiration is still done even if the rK39 RDT is positive [6]. This is potentially due to the known limitations of the rK39 RDT, which is often positive for asymptomatic cases, cannot differentiate between active or past infections, suffers from cross-reactivity with other pathogens and has shown low sensitivity in East-Africa [7]. The sensitivity of microscopic examination of tissue smears is also suboptimal because of several reasons, including improper sampling and staining, poor laboratory expertise, use of old and low resolution microscopes, and the need for relatively high parasitemia to detect parasites [8].

Due to the disadvantages of the tests that are currently used in clinical practice, there might be under- and overtreatment of patients. Therefore, there is a shift of interest towards use of molecular methods for *L. donovani* detection. Although PCR is considered most accurate to diagnose patients [9], it is costly. Solid evidence on the benefits of PCR over conventional methods for diagnosis of VL patients and its relevance for particular clinical subgroups, is lacking.

In this study, we established the proportion of suspected VL patients with a microscopically negative tissue smear sample who were PCR positive. Moreover, we demonstrate the proportion of patients that was left untreated, even though PCR was positive. This information can be included in to challenge the current diagnostic algorithm and implement molecular tests–in a certain clinical subgroup–as a complementary tool for appropriate allocation of treatment to VL patients.

## Methods

### Ethical considerations

The study was approved by the Ethical review committee of the Institute of Biotechnology, Department of Medical Biotechnology, University of Gondar (UoG), IoB/908/05/2020.

### Study site

The study was carried out at the Leishmaniasis Research and Treatment Center (LRTC) at the UoG Hospital, founded by the Drugs for Neglected Diseases initiative (DND*i*), in northwest Ethiopia. The center provides free diagnostic (rK39 RDT and microscopy) and treatment services for *leishmaniasis* patients and has a fully established molecular laboratory with trained personnel, which is currently only being used for research purposes. This laboratory is "good clinical laboratory practice" (GCLP-) compliant and twice per year internal and external quality controls are conducted for *Leishmania* detection using PCR.

When parasitological confirmation by microscopy is requested by the physician to diagnose the patient, a splenic aspirate is preferred because of its better yield. However, if the patient has an increased risk of bleeding, small or non-palpable spleen, presence of ascites or is pregnant, a bone marrow aspirate will be used. While rK39 tests will only be employed for primary VL patients, microscopy will be performed for primary VL cases, relapsing patients (symptoms within 6 months after cure) and for HIV patients or patients with a poor clinical treatment response as a test-of-cure (TOC). HIV tests are routinely performed for all primary VL patients of who the HIV status is not yet known.

### Study design

We conducted a retrospective study on stored, Giemsa-stained spleen and bone marrow smears from suspected *(i)* primary VL patients', *(ii)* relapse cases' and *(iii)* TOC samples collected at the LRTC between June 2019 and November 2020. All negative slides, except the ones that were damaged or broken, and smears with insufficient sample (less than a fifth of the slide

covered with tissue), were included. Additionally, a random selection of 14 positive slides that were behind each other with varying parasite gradings (two times +1, six times +2, three times +3, and one of +4, +5 and +6) was collected to demonstrate if there is a difference in Ct values compared to the negative slides.

## Patient data collection

For each sample included in the study, the following information about the patient was collected: socio-demographic information (age, sex, travel history) clinical parameters (fever, duration of symptoms, splenomegaly, weight loss) and laboratory parameters (HIV test results, white blood cell (WBC), hemoglobin and platelet counts, microscopy grading). Only from primary VL suspected cases, results of the rK39 RDT were additionally recorded.

## Microscopy

The microscopy slides that were assembled for this study were prepared as follows. After tissue aspiration, the samples were immediately put onto a clean glass slide to prepare a thin film smear. The smears were air dried, fixed with methanol and Giemsa staining was performed. The slides were examined systemically using 100x magnification and the parasite load was reported according to WHO guidelines (graded from 0 to +6). After microscopic examination, all slides were stored at room temperature in a slide box until use for this study. Empty slides were stored in between the other microscopically negative slides as a control for contamination between slides.

## DNA extraction

DNA was extracted from stained spleen or bone marrow smear slides and five empty control slides using the Maxwell 16 LEV Blood DNA extraction kit (Promega, Leiden, The Netherlands). In order to collect the sample from the microscopy slide, 30 μL of lysis buffer was dropped onto half of the smear (or the whole smear in case there was insufficient tissue) and the material was scraped off and added to final volume of 300 μL lysis buffer. A negative extraction control (NEC, only lysis buffer) was used for each extraction batch of 15 samples. After a brief vortex (3 sec), 30 μL proteinase K was added and the sample was incubated at $56^{\circ}$C at 400 rpm for 20 minutes. Then the samples were transferred into the automated Maxwell 16 Instrument (AS1000, Promega), which was set-up according to the manufacturer's instructions. Finally, the DNA was eluted in 50 μL elution buffer and stored at $-80^{\circ}$C until further analysis.

## *Leishmania* detection

*Leishmania* DNA detection was performed using a real-time PCR targeting the minicircle kinetoplast DNA (kDNA) with primers adopted from Mary *et al.* [10] as described before [11]. Briefly, a 25μL reaction volume was prepared with 1×HotStarTaq Master mix (Qiagen, Venlo, The Netherlands), 0.6 μM of both primers kDNA–CMF (CTTTTCTGGTCCTCCGGGTAGG, Integrated DNA Technologies (IDT), Leuven, Belgium) and kDNA–CMR (CCACCCGGCC CTATTTTACACCAA, IDT), 0.4μM of the probe kDNA–CMP (56-FAM–TTTTCGCAG / ZEN / AACGCCCCTACCCGC / 3IABkFQ, IDT), 0.1 mg/mL BSA (Roche, Vilvoorde, Belgium) and 5 μL DNA template.

In each run, a positive PCR control (*L. donovani*, Sidon_LG12_2, 100 pg/reaction) was used in duplicate to evaluate the PCR performance, and two negative PCR controls (nuclease free water and elution buffer) and the NEC were included to check for contamination. The

PCR was run on a Rotor-Gene Q instrument (Qiagen) with the following cycling conditions: 15 min at 95˚C for initial denaturation, followed by 50 cycles of 5 sec at 95˚C, 20 sec at 58˚C and 30 sec at 72˚C.

A sample was tested once in each PCR run and positive results were expressed semi-quantitatively with cycle threshold (Ct)-values. When the Ct-value was under 35, the sample was considered positive. Whenever a sample had a high Ct-value (Ct ≥ 35), the sample was re-tested in the next run to confirm the positive result. If the sample again provided a signal with any Ct-value, it was considered positive. If the sample was negative (no fluorescence) during retesting, the final result was negative.

All samples that were negative, were checked for PCR inhibition, extraction efficiency and sample sufficiency by a qPCR targeting the hemoglobin subunit beta (HBB) gene, based on primers developed by Steinau *et al.* [12] and a new probe. Briefly, a 25 µL reaction mix was made, consisting of 0.2 µM of both primers HBB-F (CAG GTA CGG CTG TCA TCA CTT AGA, IDT) and HBB-R (CAT GGT GTC TGT TTG AGG TTG CTA, IDT), 0.4 µM probe HBB-P (5TexRd-XN / TGC CCT CCC TGC TCC TGG GA / 3IAbRQSp, IDT), 1x HotStarTaq Master Mix, 0.1 mg/ml BSA and 5 µL of DNA. Cycling conditions consisted of an initial activation for 15 min at 95˚C, followed by 50 cycles of 5 sec at 95˚C, 20 sec at 58˚C and 30 sec at 72˚C. In case samples were negative for HBB, the result was considered invalid and samples were excluded from the analysis.

### Statistical analysis

A classic unpaired t-test was used to compare the obtained Ct values of microscopy negative and positive smears. A backwards logistic regression was performed on the microscopy negative smears to assess the potential associated factors for PCR positivity, in which only variables with p-values below 0.1 were left in the model. Predictors considered were: sample stage (primary VL/TOC/relapse), sample type (bone marrow/spleen aspiration), age (years), fever (yes/no and in ˚C), splenomegaly (yes/no and in cm, measured as the enlargement of the spleen compared to a normal spleen), log WBC count, RBC count, log platelet count, HIV (yes/no/missing), symptom duration (log of days), weight loss (yes/no) and travel history (yes/no). Sex was not considered as the vast majority of patients was male. Sensitivity and specificity were defined as the proportion of correctly predicted cases against PCR as reference, and at the cut-off of 50% probability to be PCR positive for the predictive model. Analyses were done using R version 4.0.3 and GraphPad version 9.4.1.

### Results

Among the 193 eligible microscopically negative slides, 57 were bone marrow and 136 spleen samples. Of these, 126 were from primary VL cases, 38 from relapse patients and 29 were TOC samples (S1 Table). Two patients were included in the study twice; one with a primary VL and TOC sample, and one with a TOC and relapse sample. Among the microscopically negative samples, two of suspected primary VL patients were not positive for HBB, indicating insufficient sample or an inefficient extraction, thus these samples were excluded from the analysis, leaving 191 microscopically negative samples.

All except five microscopically negative slides included in the study were derived from male patients and patients were generally young adults (median age 26, IQR 22–32). Every patient had symptoms for more than two weeks, 153 (79.3%) had fever, 186 (96.4%) had splenomegaly, and 130 (67.4%) indicated weight loss. The HIV status was known for 129 patients of which 13 (10.1%) were positive. Leukopenia, anemia and thrombocytopenia occurred in 131

(67.9%, lower limit 3200 WBC/μL), 147 (76.2%, lower limit male; female 11.5;11.0 g/dL) and 152 (78.8%, lower limit 128,000 plt/μL) patients, respectively [13,14].

## PCR positivity in microscopically negative slides

Among the 191 patients who had a negative microscopy result, 119 (62.3%) were positive by PCR (Fig 1, right, S2 Table). In particular, 75 out of 124 (60.5%) primary VL cases remained undiagnosed by microscopy but were positive for PCR. More than three quarters of these patients (*n* = 58; 77.3%) had not received antileishmanial treatment, yet for 39 of them (67.2%) rK39 RDT was also positive. A similar proportion of the 38 relapse patients' samples that were microscopically negative were positive by PCR (*n* = 23, 60.5%) and for 16 (69.6%) of them treatment was not reinitiated (Fig 1, right).

After referral, for some microscopically negative, non-treated, PCR positive suspected primary VL and relapse patients a differential diagnosis was available. Twenty-seven were diagnosed with malaria, 6 with viral hepatitis, 4 with pneumonia, 3 with TB and 1 with typhoid fever. Clinical outcomes of these patients were not available from the charts.

Treatment was provided, however, for 6 (12.2%) of the 49 primary VL patients and 3 (20.0%) of the 15 relapse patients who had both a negative microscopy and PCR result (Fig 1, left). Five primary VL patients were treated with only a positive rK39 RDT result and four patients were treated without any positive test (1 primary VL and 3 relapse patients, S3 Table), only based on the case definition. For the six patients from which an outcome was available, all were clinically cured.

Although PCR identified a considerable number of additional *Leishmania* DNA positive samples for both sample types, 46 of the 56 (82.1%) microscopically negative bone marrow samples turned out PCR positive, which is considerably higher compared to the microscopically negative spleen samples, for which 73 out of 135 (54.1%) were PCR positive (Fig 2). The proportion of primary VL and relapse patients that had a positive PCR test and were treated

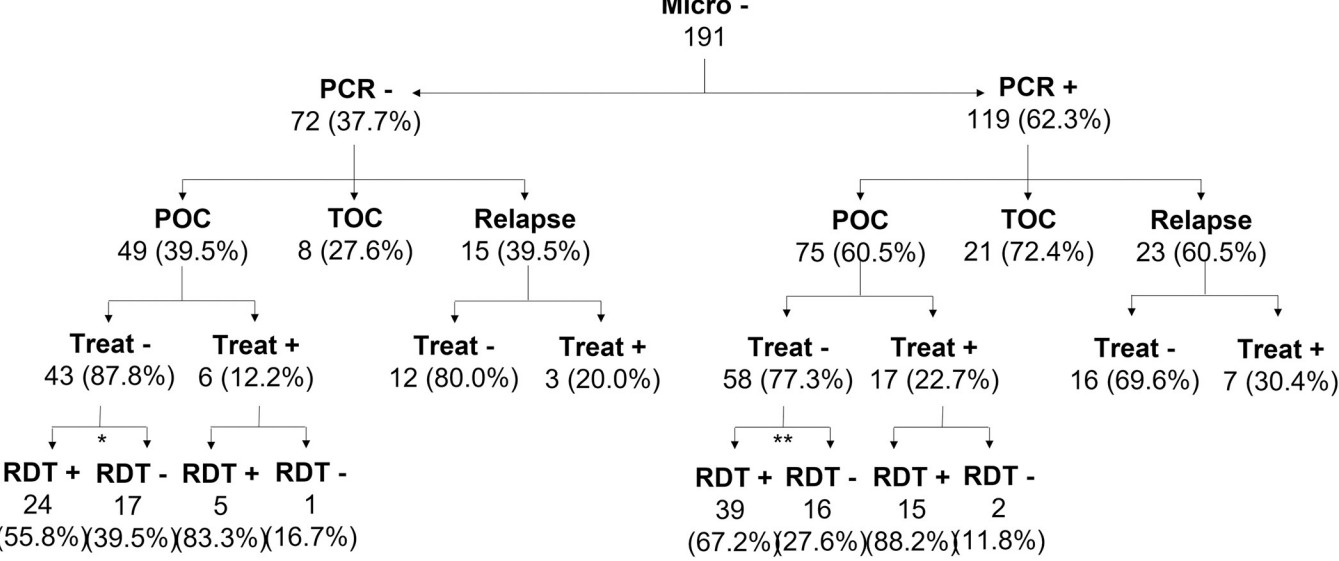

**Fig 1. PCR results and treatment provided to patients with a negative microscopy test.** The denominator for calculation of the proportion of POC, TOC and relapse samples that were PCR positive, was the total number of samples within that sampling stage; *2 patients (4.1%) not tested by rK39, **3 patients (4.0%) not tested by rK39. *Abbreviations*: Micro, microscopy; POC, point of care/primary VL patients; TOC, test of cure after antileishmanial treatment, *Treat*, treatment provided or not; RDT, rK39 rapid diagnostic test positive or negative.

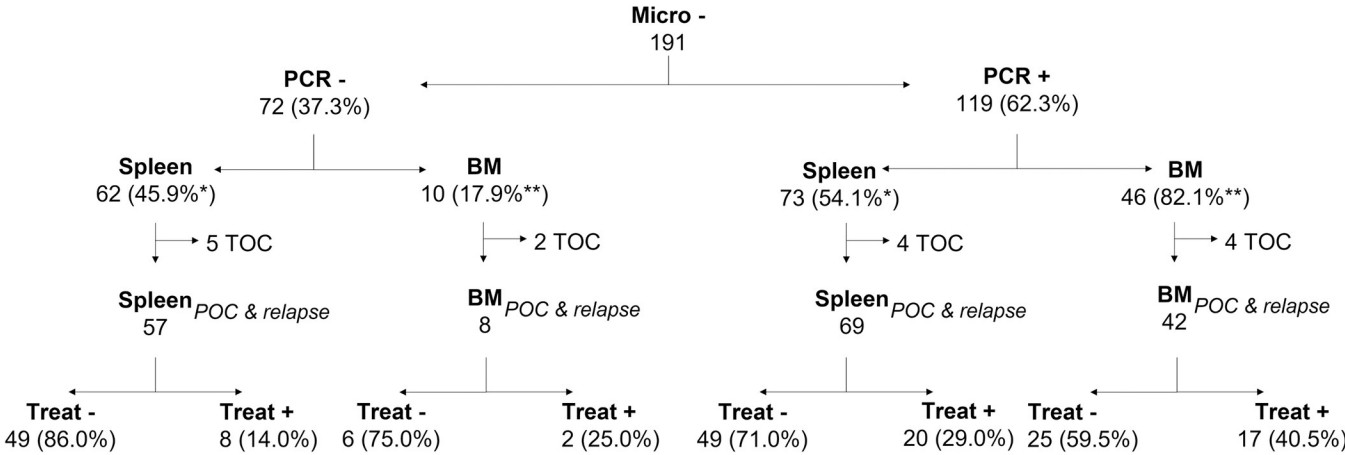

**Fig 2. PCR results and treatment provided to patients with a negative bone marrow and splenic aspirate microscopy test.** The microscope depicts the microscopy result; the instrument the PCR result, the spleen symbol the microscopy slides containing splenic aspirates; the bone symbol the microscopy slides containing bone marrow aspirates; the syringe whether or not patients were treated. *denominator is all the microscopically negative splenic aspirates, **denominator is all microscopically negative bone marrow aspirates. *Abbreviations: Micro, microscopy; BM, bone marrow sample; POC, point of care/primary VL patients; TOC, test of cure after antileishmanial treatment; Treat, treatment provided or not.*

was higher among the microscopically negative bone marrow samples (17/42, 40.5%) compared to the splenic aspirates (20/69, 29.0%).

One out of 14 microscopy positive samples was negative for PCR and rK39, but had a splenic aspirate with a microscopic grading of +1. This primary VL patient had no HIV, symptoms for one month, fever, splenomegaly (10 cm), travel history to a VL endemic area and anemia (5.8 g/μL) but WBC and platelet counts were within range. The patient received antileishmanial treatment and cured.

Overall, the rK39 RDT was positive for 54 out of 75 (72.0%, 3 patients not tested) and 29 of the 49 (59.2%, 2 patients not tested) of the primary VL patients with a PCR positive and negative result respectively.

## Ct values of microscopy positive and negative slides

Fig 3 shows that the Ct values obtained by the kDNA PCR for microscopically negative slides (median 33.0, IQR 27.7–35.9) overlapped, but were significantly higher ($p < 0.001$) than the Ct values of microscopy positive slides (median 19.6, IQR 18.0–21.2). All negative control microscopy slides that were tested to check for contamination among slides were negative by PCR.

## Prediction of the PCR results with clinical parameters

The prediction model included the variables sample type (bone marrow/spleen), log of WBC count and splenomegaly (cm), had a mean squared error of 0.211 and AUC of 0.685, showing moderate discrimination between PCR positives and negatives based on other measurements. The estimated odds of being PCR positive was considerably higher for bone marrow compared to splenic aspirates (OR = 3.87, 95% CI: 1.84–8.85) and was higher for lower white blood cell values (at half the cell count, OR = 1.56, 95% CI: 1.00–2.50). Additionally, the odds of being PCR positive was slightly higher for patients with an enlarged spleen size (OR = 1.06, 95% CI 1.00–1.12 for each difference of one cm), although only seven patients had no splenomegaly.

Using a cut-off of 50% probability to be predicted positive, the sensitivity of the prediction rule in this sample population was 81% (95% CI: 72%– 87%) and the specificity 39% (95% CI:

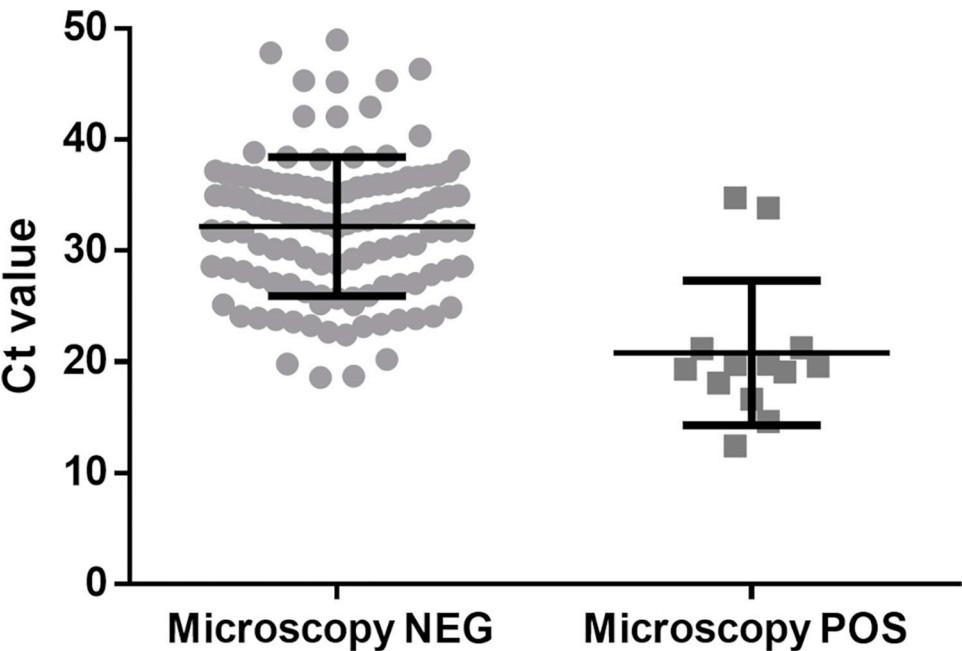

**Fig 3. Kinetoplast DNA PCR cycle threshold (Ct) values of microscopically negative (*n* = 191) and positive (*n* = 14) smears.** The unpaired t-test showed that Ct values are significantly different (p < 0.001).

28%– 51%), as shown in Table 1. This implies that the model with clinical parameters is not accurate enough for prediction and cannot replace the value of PCR. However, it can potentially be useful to prioritise which patients and samples to include for PCR testing in clinical practice when capacity is limited. Overall, patients with a microscopy negative sample have at least 30% probability to be PCR positive, and patients with a microscopically negative bone barrow sample even minimum 50%.

## Discussion

Our research demonstrates that more than half of the microscopically negative tissue smear slides of suspected patients were positive by PCR. When VL cases remain undiagnosed, they have a high risk of dying and can further contribute as a reservoir to transmission. Accordingly, if East-Africa wants to move towards VL elimination, there is an urgent need for better diagnostics.

For microscopically negative bone marrow aspirates, the PCR positive proportion was considerably higher than among splenic aspirates, reaching up to 80% PCR positivity. Bone marrow aspirates are known to be less sensitive for diagnosis of VL by microscopy, hence they are only used if splenic aspiration is not advisable due to a risk of bleeding [5,15]. It is the sicker patients (tendency to bleed, low platelet counts, ascites) who undergo bone marrow aspiration, for whom the parasite loads are expected to be higher [6]. However, because the volume of the sample is larger, it is presumably more difficult to find the parasites under the microscope.

**Table 1. Sensitivity and specificity of the prediction model, with samples above the cut-off of 50% probability to be positive according to the model considered predicted positive.**

|              | Predicted negative | Predicted positive |
|--------------|--------------------|--------------------|
| PCR negative | 28                 | 44                 |
| PCR positive | 23                 | 96                 |

PCR amplifies the high copy number kDNA fragment, and is therefore able to detect DNA of low parasite loads which might not be visible under the microscope [10]. The high proportion of PCR positive slides is in line with another study performed in Brazil on a smaller sample set, which found 16 out of 19 (84.2%) microscopically negative bone marrow slides of suspected VL patients positive by PCR [16]. In contrast, a study in Nepal found only 43.5% of the 50 microscopically negative bone marrow samples PCR positive [17]. However, the assay used for *Leishmania* detection was less sensitive than the one used in our study and storage time of these samples was long, which could have caused decay of the DNA [18]. The PCR positivity among the test of cure samples was slightly higher compared to primary VL and relapse cases. It should be taken into account that even though kDNA decay is described to be quite quick, there is presumably still kDNA detection when parasites are not viable anymore or cleared. A PCR that can detect only viable *Leishmania* parasites, like the spliced leader (SL-) RNA qPCR could be an added value as a test-of-cure [19], although its sensitivity is slightly lower than the kDNA qPCR [20,21].

Importantly, we found that almost three quarters of the suspected primary VL patients with a negative microscopy but positive PCR result did not receive antileishmanial treatment, which could be fatal for the patients. Although approximately all patients in our study had symptoms already for more than 2 weeks, they were potentially still in an early phase of the disease with low parasite loads, hampering parasite detection by microscopy and treatment accordingly. However, interpretation of the PCR results should be done with care before deciding to treat all PCR positive patients. First, even though control slides were stored between the other slides, we can never completely rule out that there was no contamination. Moreover, the Ct values of the microscopically negative slides were in general higher than the microscopically positive slides, meaning that there were less parasite equivalents detected in the microscopically negative samples. PCR positivity is also used to identify asymptomatic patients, who are not treated due to the toxicity of the antileishmanial drugs [11,22]. Therefore, case-by-case discussions should be organized between the treating physician and molecular biologist to evaluate whether or not to treat a patient with a positive PCR test. If decided not to treat the patient, they should be closely monitored to assess their disease progression and return to the treatment center in case the symptoms are worsening.

Almost 30% of the patients that were microscopically negative, yet PCR positive and did not receive treatment had a differential diagnosis of malaria (determined by microscopy) and were treated accordingly. These patients could potentially have a coinfection, although a study among migrant workers in northwest Ethiopia showed that the malaria coinfection rate was only 2.8% [23].

Some patients in this study were treated based only on clinical suspicion and a positive rK39 RDT result, which was not confirmed by PCR. The patients all had a good clinical outcome, which could indicate they did truly have VL and were treated appropriately with a low-quality sample or low parasite load leading to a false negative microscopy and PCR result; or alternatively, they could be true negative cases and not have had VL at all. These patients were potentially overtreated, which should be avoided due to the toxicity of the treatment. The proportion of patients that were PCR positive and did receive treatment, even though the microscopy results were negative, was higher among patients of whom a bone marrow sample was collected, indicating that clinicians may keep the lower sensitivity of bone marrow aspirates into account in their decision to treat empirically.

The rK39 RDT confirmed all microscopy and a considerable amount of the PCR positive results for suspected primary VL patients, which could indicate that its diagnostic performance might be better than previously reported in East-Africa [24,25]. However, the RDT was also positive for many unconfirmed samples. This is presumably due to the fact that most patients

live in or have a travel history to VL endemic areas around Gondar [26,27] which could have led to a previous (asymptomatic) infection. However, it should be considered that the rK39 RDT positivity rate was overall very high. An important reason for this is that previous research has shown that the decision to conduct a tissue biopsy was associated with a positive rK39 RDT result, creating a bias in the sampled population [6,24].

We found one sample that was microscopically positive but negative by PCR, which can be due to several reasons. First, since the sample was graded +1, it could be that the parasites were not equally distributed on the slide, and as only part of the tissue on the slide was used for extraction, the parasites may not have been present in that part of the slide. Second, there could have been a staining or reading error. Third, the storage time until extraction could have played a role, which can have led to *Leishmania* DNA degradation. Although all samples were extracted within a year after collection, other similar studies on archived microscopically positive slides found that the older slides result in reduced PCR positivity [18]. If PCR would be included in the diagnostic algorithm, this issue would not occur as tissue aspirates would be extracted from the slides for PCR immediately after a negative microscopy result.

Overall, implementation of molecular tools in the diagnostic algorithm for VL could have a large impact on patient care. In this retrospective study we were only able to collect 191 negative slides in about 1.5 years time, as negative slides were not always (properly) stored. Because it is currently not routinely done, some slides were dusty, broken or not clearly labeled and therefore excluded for the study. If this would be done properly, many additional slides could have been included. In 2020, 1,065 suspected VL patients were referred to the LRTC in Gondar and tissue aspiration was done for 619 of them. Only 187 (30.2%) were microscopically positive, leaving yearly about 432 patients unconfirmed. Based on results from our study, showing that more than half of these could be PCR positive, implementation of PCR could identify more than 200 additional VL patients yearly in a single treatment site. However, an important limitation of the study is the fact that samples that did not have sufficient tissue were not used for DNA isolation and PCR. As splenic aspirates are usually small in volume, presumably less splenic aspirate smears have been included in our study (although these data were not recorded). Including them could either have led to more PCR positives among the splenic aspirates, if the tissue was still sufficient for PCR; or less PCR positives, if the amount of tissue was also insufficient for PCR. If further studies are done, these samples should not be excluded for analysis.

Although PCR was proven very useful to detect VL, it cannot be implemented as a first-line diagnostic tool in a resource-limited setting like Ethiopia as it is costly and rapid diagnosis is needed so that the patient can be treated. This is unfortunate, as the model indicated that PCR positivity cannot be explained with or predicted by routinely available clinical data. However, as slides are easy to transport, negative slides could be sent from primary health care facilities where microscopic diagnosis is conducted to referral hospitals where they can be tested by PCR. If resources are restricted, one could further develop a diagnostic algorithm, including the findings of the prediction model. Such algorithm would prioritize bone marrow samples, and patients with splenomegaly and especially low WBC values to be tested as they have a higher chance of being PCR positive. Alternatively, an affordable and simple test (e.g., loop-mediated isothermal amplification [28] or a novel RDT) on a non-invasive sample like peripheral blood would be very useful to include in the diagnostic algorithm in VL endemic areas for early diagnosis and treatment of VL patients.

## Conclusions

Collectively, this study demonstrates that more than half of the tissue slides of suspected VL patients which are negative by microscopy were positive by PCR, indicating the urgency for

better diagnostic tools in Ethiopia. Almost three quarters of the patients that were PCR positive were not treated with antileishmanial drugs. How such patients' disease progresses is not well understood, but close monitoring is advised. Overall, the current diagnostic algorithm should be challenged and molecular tools should be implemented, at least on microscopically negative bone marrow aspirate smears. This could lead to considerable improvements in patient management in Ethiopia, and beyond.

## Supporting information

**S1 Table. Overview of microscopically negative bone marrow and splenic aspirate samples from primary VL, test-of-cure (TOC), and relapse patients.** * A bone marrow and splenic aspirate sample, both from a primary VL patient were not included in the table, as they were not positive by the HBB PCR, indicating insufficient sample or an inefficient extraction, hence they were excluded from the study.
(DOCX)

**S2 Table. Number and percentage of PCR positive and negative results among microscopically negative tissue slides, according to sample stage and type.**
(DOCX)

**S3 Table. Overview of PCR negative patients that were treated with antileishmanial drugs.** POC: point of care, WBC: white blood cell, Hb: hemoglobin, Plt: platelets, N.A.: data not available or not found on the patient chart.
(DOCX)

## Acknowledgments

We would like to thank Lieselotte Cnops for her contribution to the design of this study and Ilse Maes for kindly providing positive controls for the PCRs. We are also grateful to the patients who agreed to be part of this study and the LRTC staff who participated in the data collection and processing.

## Author Contributions

**Conceptualization:** Roma Melkamu, Rezika Mohammed, Saskia van Henten, Johan van Griensven.

**Data curation:** Roma Melkamu, Bart K. M. Jacobs, Helina Fikre, Myrthe Pareyn.

**Formal analysis:** Roma Melkamu, Bart K. M. Jacobs, Mekibib Kassa, Arega Yeshanew, Myrthe Pareyn.

**Funding acquisition:** Johan van Griensven.

**Investigation:** Roma Melkamu, Mekibib Kassa, Helina Fikre, Saba Atnafu, Myrthe Pareyn.

**Methodology:** Roma Melkamu, Nega Berhane, Bart K. M. Jacobs, Rezika Mohammed, Arega Yeshanew, Johan van Griensven, Myrthe Pareyn.

**Project administration:** Roma Melkamu, Nega Berhane, Rezika Mohammed, Arega Yeshanew, Johan van Griensven.

**Resources:** Rezika Mohammed, Arega Yeshanew, Helina Fikre, Saba Atnafu, Johan van Griensven.

**Software:** Bart K. M. Jacobs, Myrthe Pareyn.

**Supervision:** Nega Berhane, Bart K. M. Jacobs, Rezika Mohammed, Mekibib Kassa, Helina Fikre, Johan van Griensven, Myrthe Pareyn.

**Validation:** Roma Melkamu, Myrthe Pareyn.

**Visualization:** Saskia van Henten, Johan van Griensven, Myrthe Pareyn.

**Writing – original draft:** Roma Melkamu, Myrthe Pareyn.

**Writing – review & editing:** Roma Melkamu, Nega Berhane, Bart K. M. Jacobs, Rezika Mohammed, Mekibib Kassa, Arega Yeshanew, Helina Fikre, Saba Atnafu, Saskia van Henten, Johan van Griensven, Myrthe Pareyn.

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
