## [Decision Letter · Decision Letter 0]

1 Dec 2022

Dear Ms. Melkamu,

Thank you very much for submitting your manuscript "PCR for detection of Leishmania donovani from microscopically negative tissue smears of suspected patients in Gondar, Ethiopia" for consideration at PLOS Neglected Tropical Diseases. As with all papers reviewed by the journal, your manuscript was reviewed by members of the editorial board and by several independent reviewers. In light of the reviews (below this email), we would like to invite the resubmission of a significantly-revised version that takes into account the reviewers' comments. 

Considering the observations pointed out by the referres, I suggest major revisions.

We cannot make any decision about publication until we have seen the revised manuscript and your response to the reviewers' comments. Your revised manuscript is also likely to be sent to reviewers for further evaluation.

Sincerely,

Claudia Ida Brodskyn

Academic Editor

Ana Rodriguez

Section Editor

Considering the observations pointed out by the referres, I suggest major revisions.

Reviewer's Responses to Questions

**Key Review Criteria Required for Acceptance?**

**Methods**

-Are the objectives of the study clearly articulated with a clear testable hypothesis stated?

-Is the study design appropriate to address the stated objectives?

-Is the population clearly described and appropriate for the hypothesis being tested?

-Is the sample size sufficient to ensure adequate power to address the hypothesis being tested?

-Were correct statistical analysis used to support conclusions?

-Are there concerns about ethical or regulatory requirements being met?

Reviewer #1: -Are the objectives of the study clearly articulated with a clear testable hypothesis stated? Yes

-Is the study design appropriate to address the stated objectives? Yes

-Is the population clearly described and appropriate for the hypothesis being tested? Yes

-Is the sample size sufficient to ensure adequate power to address the hypothesis being tested? yes

-Were correct statistical analysis used to support conclusions? Yes

-Are there concerns about ethical or regulatory requirements being met? Yes

Reviewer #2: (No Response)

Reviewer #3: - Line 104-105: “random selection …varying parasite gradings…” Please mention what the parasite grades were and how these were randomly picked. 

- Was the sample size of 14 from microscopy positive adequate to demonstrate the difference in Ct values?

- Line 228: define severity of splenomegaly

**Results**

-Does the analysis presented match the analysis plan?

-Are the results clearly and completely presented?

-Are the figures (Tables, Images) of sufficient quality for clarity?

Reviewer #1: -Does the analysis presented match the analysis plan? Yes

-Are the results clearly and completely presented? Yes

-Are the figures (Tables, Images) of sufficient quality for clarity? Yes

Reviewer #2: (No Response)

Reviewer #3: - It is mentioned that in 2020, 619 patients had tissue aspiration at the study site and 432 were negative. It will be clearer if a flow diagram of the study population, inclusion-exclusion is included in the paper. 

- Make sure that the + and – signs be clearly visible in the figures 1 and 2

- Line 220: the sample size for microscopy positive and microscopy negative are exchanged 

- Line 231: “At a cut-off of 50% (log-odds = 0), the sensitivity in this sample population 231 was 81%...” this statement is not clear. Please explain which variable the cut-off of 50% refers to. It is important to describe this statistical analysis in the methods section too.

**Conclusions**

-Are the conclusions supported by the data presented?

-Are the limitations of analysis clearly described?

-Do the authors discuss how these data can be helpful to advance our understanding of the topic under study?

-Is public health relevance addressed?

Reviewer #1: -Are the conclusions supported by the data presented? Yes

-Are the limitations of analysis clearly described? yes

-Do the authors discuss how these data can be helpful to advance our understanding of the topic under study? Yes

-Is public health relevance addressed? yes

Reviewer #2: (No Response)

Reviewer #3: - PCP positivity of ToC patients (72.4%) is higher than all the other microscopy negative groups (primary and relapse). It is important to discuss this finding. Is it possible that PCR is detecting non-viable DNA? 

- Higher detection of VL from bone marrow smears than spleen was found in this study. Several reasons can be entertained for this finding and may be discussed. For example, it is the sicker patient who undergo bone marrow aspiration than spleen aspiration (those with bleeding, very low platelet count, ascitis). Such patients tend to have higher parasite load. This may not be reflected on parasite grading which is typically based on spleen aspiration. Bone marrow samples tend to be more in volume and diluted than spleen aspirates. Higher chance to miss on microscopy due to diluted bone marrow sample; and scanty tissue from spleen aspirate and probably inadequate sample for PCR from the spleen slides.

**Editorial and Data Presentation Modifications?**

Reviewer #1: It would be better if author clearly define the number of spleen and bone marrow samples for each individual cohort (POC, relapse and TOC) as supplemental figure. 

Table 1: author needs to clearly define the distribution of number of cases (PCR negative) in this table in accordance with figure-1.

Reviewer #2: (No Response)

Reviewer #3: - Line 95: “small on non-palpable” – should read “small or non-palpable”

**Summary and General Comments**

Reviewer #1: This is a very good study that could make significant impact on diagnosis and care in a resource limited settings of Ethiopia. In this study author represents three critical aspects:

1) Lack of sensitivity of current diagnostic (rk39 rapid diagnostic test and microscopy of tissue smears) procedure in 

 Gondar, Ethiopia.

2) Potential of real time PCR for detection of L. donovani from microscopically negative spleen and bone marrow smears 

 of suspected patients in Gondar, Ethiopia.

3) Implementation of PCR could improve patient diagnosis and treatment at least for microscopically negative bone 

 marrow cases.

But, there are some minor issues that deserve discussion:

1) Figure-2: author represents the percentage and number of microscopically negative PCR positive bone marrow 

 samples, not correlates with the statements in the line number 28 and 196.

2) Figure-3: Number of microscopically negative and positive samples (n) in the figure, not correlate with the figure 

 legends- and with line number-104.

3) Line 211-212 – author needs to check the number of not tested patients, not correlates with the figure-1 legend.

4) Author need to clearly define how they measure sensitivity and specificity.

5) It would be better if author had measured the parasite load and show correlation between Ct 

 values of PCR positive microscopically negative samples with parasite load at least for bone marrow samples.

6) Leishmania detection: Author need to define the conc. of DNA template used for PCR study along with the volume 

 used.

According to me, this manuscript could be accepted after such changes.

Reviewer #2: Major Points

1) What was the final elute volume

2) How did you decide if there was 'insufficient material' when extracting the samples (line 125)

3) Please clarify 

" Whenever a sample had a high Ct-value (Ct ≥ 35), the sample was re-tested to confirm the positive result. If the sample was negative during retesting, the final result was negative."

How was negative defined here? For example if I ran a sample in duplicate n the first run and got CT 36/36 - this was then re-run in duplicate again? If it amplified at 37/37 again was this positive or negative? Presumably you had a CT threshold for positive/negative.

4) The statistical analysis is not adequately described in particular outlining which variables were used in the model and how categorical varibales were constructed. Equally there are comparisons provided in the results (yield of PCR between bone marrow and splenic samples for example) which were not desribed in the methods.

The results of the model are also not well presented. There is no sense of what variables were included with only a final model given.

5) I found the results quite difficult to follow mixed between the flow charts and text. Please consider revisiting this as its important information. I would strongly encourage a table stratified by status/sample type etc.

Minor Points:

Line 70-72 - "The sensitivity of microscopic examination of tissue smears is also suboptimal ......" this sentence should be referenced.

Line 127 - 'after a quick spin'. Please clarify if you literally just mean vortexing or if you mean X minutes at Y RPM so that the method could in theory be replicated.

Reviewer #3: General comments:

- The study demonstrated that a high proportion of patients were missed when RDT and tissue aspirate microscopy are used. The use of PCR increased case detection. It was reported before that the diagnostic performance of rK39 RDT in east Africa is low. This finding suggests the urgent and strong need for better diagnostics for VL in the region. This needs to be emphasized in the paper.

- Authors showed that several rK39 RDT positive cases but tissue aspirate microscopy were positive on PCR testing. It is important to discuss if this might show that rK39 RDT actually have better performance than reported; and that lower rK39RDT performance might be due to reference test bias (low sensitivity of tissue aspirate microscopy). 

- It this retrospective study at a busy leishmaniasis treatment and research center, there can be easy contamination of samples during processing or storage of slides. It is important to describe what measures are place to avoid that. Is keeping empty slides between tissue smear slides a routine activity? Contamination of samples (slides) can have significant impact on the findings of such study (molecular diagnostics) and needs to be described clearly or discussed as a potential limitation. 

Abstract

- Line 25: The number and proportion of bone marrow and spleen aspirate may be included with this sentence mentioning the total sample size.

- More strong recommendation on the need for better diagnostics for VL in the region can be provided here given the significant gaps demonstrated. 

Author summary

- Line 50: recommends implementing PCR at least on microscopically negative bone marrow aspirate smears. This sentence undermines the fact that fewer proportion of patient undergo bone marrow aspiration, and still the PCR positivity on spleen aspiration microscopy negative high. 

Introduction

- Line 59-60: States the importance of reliable diagnostic methods for treatment decision making. The prior sentences describe the magnitude of VL and its fatality. It is important, additionally, to mention the cost implications and adverse effects related to treatment. The need for reliable diagnostics is due to the fact that the complication of the disease needs to be balanced with the complication of the treatment.

PLOS authors have the option to publish the peer review history of their article (what does this mean?). If published, this will include your full peer review and any attached files.

Reviewer #1: No

Reviewer #2: No

Reviewer #3: No
---

## [Decision Letter · Decision Letter 1]

20 Jan 2023

Dear Ms. Melkamu,

Thank you very much for submitting your manuscript "PCR for detection of Leishmania donovani from microscopically negative tissue smears of suspected patients in Gondar, Ethiopia" for consideration at PLOS Neglected Tropical Diseases. As with all papers reviewed by the journal, your manuscript was reviewed by members of the editorial board and by several independent reviewers. The reviewers appreciated the attention to an important topic. Based on the reviews, we are likely to accept this manuscript for publication, providing that you modify the manuscript according to the review recommendations. 

We accept the manuscript, but i strongly recommend to make the modification suggested by the reviewer inn order to improve the manuscript:

In response to reviewer 3 you have said

"The positive slides were collected through random sampling of positive slides at convenience. "

Random sampling and convenience sampling are fundamentally different. Was random sampling or convenience sampling used.

- Line 228: define severity of splenomegaly

The sentence in the manuscript was revised to “patients with a larger spleen size” to clarify the

severity of splenomegaly.

This doesnt really provide clarification. I think you need to either give a definition of larger or state more clearly tht you fitted spleen size as a linear variable - if so did you assess for evidence of non-lineatiy of the associations?

Sincerely,

Claudia Ida Brodskyn

Academic Editor

Ana Rodriguez

Section Editor

We accept the manuscript, but i strongly recommend to make the modification suggested by the reviewer inn order to improve the manuscript:

In response to reviewer 3 you have said

"The positive slides were collected through random sampling of positive slides at convenience. "

Random sampling and convenience sampling are fundamentally different. Was random sampling or convenience sampling used.

- Line 228: define severity of splenomegaly

The sentence in the manuscript was revised to “patients with a larger spleen size” to clarify the

severity of splenomegaly.

This doesnt really provide clarification. I think you need to either give a definition of larger or state more clearly tht you fitted spleen size as a linear variable - if so did you assess for evidence of non-lineatiy of the associations?

Reviewer's Responses to Questions

**Key Review Criteria Required for Acceptance?**

**Methods**

-Are the objectives of the study clearly articulated with a clear testable hypothesis stated?

-Is the study design appropriate to address the stated objectives?

-Is the population clearly described and appropriate for the hypothesis being tested?

-Is the sample size sufficient to ensure adequate power to address the hypothesis being tested?

-Were correct statistical analysis used to support conclusions?

-Are there concerns about ethical or regulatory requirements being met?

Reviewer #2: (No Response)

Reviewer #3: (No Response)

**Results**

-Does the analysis presented match the analysis plan?

-Are the results clearly and completely presented?

-Are the figures (Tables, Images) of sufficient quality for clarity?

Reviewer #2: (No Response)

Reviewer #3: (No Response)

**Conclusions**

-Are the conclusions supported by the data presented?

-Are the limitations of analysis clearly described?

-Do the authors discuss how these data can be helpful to advance our understanding of the topic under study?

-Is public health relevance addressed?

Reviewer #2: (No Response)

Reviewer #3: (No Response)

**Editorial and Data Presentation Modifications?**

Reviewer #2: (No Response)

Reviewer #3: (No Response)

**Summary and General Comments**

Reviewer #2: In response to reviewer 3 you have said 

"The positive slides were collected through random sampling of positive slides at convenience. "

Random sampling and convenience sampling are fundamentally different. Was random sampling or convenience sampling used.

- Line 228: define severity of splenomegaly

The sentence in the manuscript was revised to “patients with a larger spleen size” to clarify the

severity of splenomegaly. 

This doesnt really provide clarification. I think you need to either give a definition of larger or state more clearly tht you fitted spleen size as a linear variable - if so did you assess for evidence of non-lineatiy of the associations?

Otherwise I am satisfied with the changes

Reviewer #3: (No Response)

PLOS authors have the option to publish the peer review history of their article (what does this mean?). If published, this will include your full peer review and any attached files.

Reviewer #2: No

Reviewer #3: No

Figure Files:

Data Requirements:

Reproducibility:

References

---

## [Editor Report · Decision Letter 2]

31 Jan 2023

Dear Ms. Melkamu,

We are pleased to inform you that your manuscript 'PCR for detection of Leishmania donovani from microscopically negative tissue smears of suspected patients in Gondar, Ethiopia' has been provisionally accepted for publication in PLOS Neglected Tropical Diseases.

Best regards,

Claudia Ida Brodskyn

Academic Editor

Ana Rodriguez

Section Editor

Dear Dr.,

After the modifications introduced in the manuscript, we accept it to be published in PNTD.

Best regards

Cláudia

---

## [Editor Report · Acceptance letter]

8 Feb 2023

Dear Ms. Melkamu,

We are delighted to inform you that your manuscript, "PCR for detection of Leishmania donovani from microscopically negative tissue smears of suspected patients in Gondar, Ethiopia," has been formally accepted for publication in PLOS Neglected Tropical Diseases.

Best regards,

Shaden Kamhawi

co-Editor-in-Chief

Paul Brindley

co-Editor-in-Chief
